# Burning Properties of Combined Glued Laminated Timber

Tomáš Kytka, Miroslav Gašparík, David Novák, Lukáš Sahula, Elham Karami * and Sumanta Das

Department of Wood Processing and Biomaterials, Faculty of Forestry and Wood Sciences, Czech University of Life Sciences Prague, Kamýcká 1176, Suchdol, 165 00 Prague, Czech Republic; kytkat@fld.czu.cz (T.K.); gasparik@fld.czu.cz (M.G.); novakdavid@fld.czu.cz (D.N.); sahulal@fld.czu.cz (L.S.); dass@fld.czu.cz (S.D.)
* Correspondence: karami@fld.czu.cz

**Abstract:** This study delved into the combustion properties of combined glulam bonded using polyurethane (PUR) and resorcinol-phenol-formaldehyde (RPF) adhesives. The experiment involved three distinct wood species, namely, spruce, alder, and beech, which were combined in homogeneous, non-homogeneous symmetrical, and non-homogeneous asymmetrical arrangements. These species were selected to represent a spectrum, namely, softwood (spruce), low-density hardwood (alder), and high-density hardwood (beech). The varying combinations of wood species illustrate potential compositions within structural elements, aiming to optimize mechanical bending resistance. Various parameters were measured during combustion, namely, the heat release rate (HRR), peak heat release rate (pHRR), mass loss rate (MLR), average rate of heat emission (ARHE), peak average rate of heat emission (MARHE), time to ignition (TTI), and effective heat of combustion (EHC). The findings indicate that incorporating beech wood into the composite glulam resulted in an increase in heat release, significantly altering the burning characteristics, which was particularly evident at the second peak. Conversely, the use of spruce wood exhibited the lowest heat release rate. Alder wood, when subjected to heat flux at the glued joint, displayed the highest heat emission, aligning with the results for EHC and MARHE. This observation suggests that wood species prone to early thermal decomposition emit more heat within a shorter duration. The time to ignition (TTI) was consistent, occurring between the first and second minute across all tested wood species and combinations. Notably, when subjected to heat flux, the glulam samples bonded with PUR adhesive experienced complete delamination of the initial two glued joints, whereas those bonded with RPF adhesive exhibited only partial delamination.

**Keywords:** combined glulam; burning; heat release rate; ignition; spruce; alder; beech





## 1. Introduction

Glued laminated timber (GLT or glulam) has become a prevalent building material. However, owing to its biological composition, wood is inherently prone to combustion [1]. When subjected to significant heat, the constituents within wood undergo pyrolysis, generating flammable gases and a charred layer. The heat transferred to the wood from an external source is crucial for initiating pyrolysis [2]. Despite carbon's excellent heat conductivity, the charred layer within wood functions as an insulator. This insulating effect primarily arises from the porous and heterogeneous structure of the charred layer, impeding heat transfer via conduction [3]. As the thickness of the charred layer increases, the flame height diminishes due to the deceleration of wood pyrolysis. Furthermore, the flame resulting from the combustion of gases emanating from the wood acts as a barrier, impeding the original heat source from transmitting heat flux to the wood [4].

The concept of the energy balance serves as a fundamental tool in comprehending the combustion dynamics of solid materials [5]. The energy inputs involved in the combustion reaction encompass the heat provided by an external source, the radiation of heat from the flame, and the heat generated from the oxidation of the charred wood layer [6]. This cumulative input is counterbalanced by the energy demanded for the gasification of wood

components within the pyrolysis region and the accumulation of energy within the charred layer. This stored energy is subsequently dissipated through radiation and within the pyrolysis region back to the original material [7]. Additional losses, such as the energy necessary for heating and evaporating water contained within the wood, also significantly influence this balance [8]. An augmentation in the thickness of the charred layer alters this equilibrium to a critical point where the wood begins to manifest a self-extinguishing effect [9].

The characteristics of wood, including its species, anatomical structure, and chemical composition, play a definitive role in its combustion behavior. Particularly, the chemical composition significantly influences the thermal properties and subsequent combustion of wood [10]. Wood comprises a complex array of chemical compounds, which are predominantly constituted of cellulose, hemicelluloses, and lignin. These three compounds serve as the foundational constituents across all wood species, exerting substantial influence over the thermal properties of wood [11]. In addition to cellulose, hemicelluloses, and lignin, wood contains smaller quantities of extractives, such as terpenes, tannins, fats, and waxes. Unlike cellulose, these extractives possess lower thermal stability and are prone to decomposition at an earlier stage during combustion among the wood constituents [12].

The thermal properties of wood, namely, thermal diffusivity, thermal conductivity, and specific heat, are pivotal factors influencing its combustion behavior. These properties are significantly influenced by several physical attributes, primarily the moisture content, temperature, wood density, porosity, and the orientation of heat flux concerning the fiber orientation within the wood [13,14]. These parameters collectively determine the barriers that the applied heat flux must surpass to initiate wood pyrolysis and commence the combustion process. Furthermore, the magnitude of these thermal properties varies uniquely for each wood species (see Table 1). Considering the extensive heterogeneity and anisotropy inherent in wood, these properties assume considerable importance in designing wood structures engineered to withstand fire events [15].

**Table 1.** Thermal properties of selected wood species.

| | Density [kg/m³] | Thermal Conductivity at 20 °C (L/T) [W/mK] | Specific Heat Capacity at 20 °C [J/kgK] | Source |
|---|---|---|---|---|
| Alder | 492 | 0.273/0.166 | 1400 | [16,17] |
| Beech | 710 | 0.515/0.127 | 1213 | [18] |
| Spruce | 471 | 0.352/0.0956 | 1244 | [18] |

The primary parameters observed during combustion include the mass loss rate (MLR), which denotes the rate at which a material sheds mass, without discerning which part of the material is consumed. For organic materials like wood, this rate is significantly affected by decreasing moisture content [19]. A higher mass loss rate often indicates a greater amount of available fuel for the fire, potentially resulting in elevated heat release rates and more intense fires. As more combustible material is consumed, the fire's intensity intensifies, resulting in larger flames, heightened temperatures, and a faster propagation of the fire to adjacent materials or areas [20]. Controlling or diminishing mass loss is crucial for fire containment strategies, emphasizing the reduction in the fire's fuel supply. This can be achieved either by directly extinguishing the fire or by implementing tactics to prevent its further spread [21]. The second pivotal variable is the heat release rate (HRR), indicating the magnitude with which the material contributes thermal energy to the combustion process [22]. A higher HRR can trigger rapid fire escalation, increased temperatures, and faster exposure of structural elements to intense heat. Consequently, this accelerated exposure may cause structural components to reach critical temperatures sooner than they would in scenarios with lower heat release rates [23].

Presently, glued laminated timber (glulam, GLT) finds widespread application in structural contexts due to its superior dimensional stability; fewer inherent defects; and consequently, enhanced strength compared with solid wood. From a fire safety perspective,

despite wood's combustibility, it retains its structural integrity during a fire event owing to the creation of an insulating charred layer [24]. Conversely, unprotected steel undergoes a rapid reduction in strength upon exposure to increased temperatures [25]. When examining homogeneous glulam beams (crafted from a single wood species), investigations into the thermal characteristics and properties influencing combustion can be approached similarly to solid wood analyses [26]. However, the combination of different wood species within a glulam assembly presents distinct challenges, primarily due to variations in the anatomical structure, physical properties, and differing thermal expansion exhibited by each wood species [27,28].

The contemporary use of wood in construction is subject to numerous regulations, with Eurocode 5 [29] standing as one of the pivotal standards that delineate its permissible applications. Each structural element, component, and fastener, as well as the overall structure, must adhere to predefined criteria encompassing integrity (E), load-bearing capacity (R), and thermal insulation (I). Specifically for timber elements, compliance depends on the alteration of properties under fire conditions and whether the entire timber element is shielded during the fire's initiation shielded during whole fire or not at all. Guided by these variables, the charring depth is determined, and the remaining cross-section, which should retain its post-fire structural strength, is ascertained by utilizing methodologies like the reduced cross-section method or the reduced properties method. These approaches aid in evaluating the residual strength of timber elements after exposure to fire effects (see Figure 1).

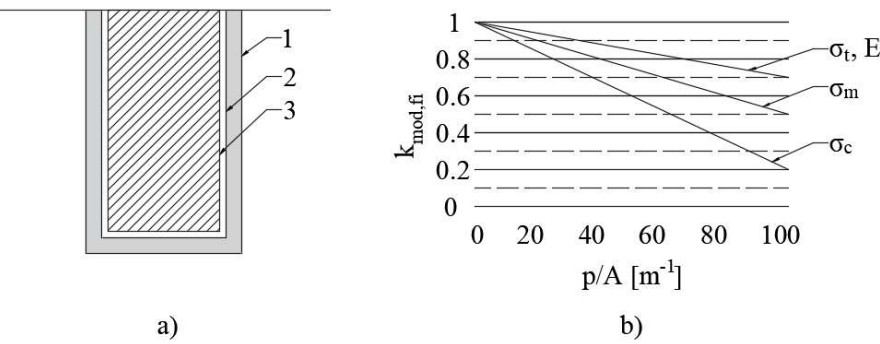

**Figure 1.** Calculation methods according to ČSN EN 1995-1-2 (2006) [29]: (**a**) reduced cross-section method and (**b**) reduced properties method (1—initial element surface, 2—edge of residual surface, 3—edge of effective cross-section, $\sigma_t$—tensile strength, $\sigma_m$—bending strength, $\sigma_c$—compressive strength, E—modulus of elasticity, $k_{mod,fi}$—modification factor for fire, p—perimeter of residual cross-section subjected to fire, A—area of residual cross-section).

In glued laminated timber, the adhesive plays a significant role as it responds to temperature fluctuations. Contemporary adhesives primarily comprise polymers derived from diverse bases, including phenolic, amine, isocyanate, polyvinyl ester, epoxy resins, and various others. Notably, the prevalent adhesive systems encompass melamine-formaldehyde (MF)- or melamine-urea-formaldehyde (MUF)-based adhesives, constituting 51% of usage, followed by polyurethane (PUR) adhesives at 35% and resorcinol (RPF) based adhesives accounting for 9% [30].

Furthermore, these adhesive bases exhibit distinct reactions to temperature variations, yet they share common critical points: the glass transition temperature and the melting temperature of the crystalline phase (the creep temperature in amorphous polymers). These temperatures mark structural and property alterations within the adhesive [31]. As the temperature increases, adhesives typically display a decline in both the modulus of elasticity and shear strength within the bonded joint. This decline primarily stems from reduced adhesion between the adhesive and the substrate, as well as decreased internal cohesion within the adhesive itself [32,33]. This behavior is notably evident in adhesives based on one-component polyurethanes, which exhibit a reduction in shear strength across

a broad temperature range (70 to 170 °C). Conversely, phenol-resorcinol-formaldehyde (PRF) resins demonstrate an initial strength reduction at approximately 180–190 °C [34]. The temperature-induced changes can lead to adhesive strength limits being surpassed, culminating in delamination of the bonded joint and potential beam failure. However, such behavior is deemed unacceptable in structural materials, particularly during fire exposure, as delamination may trigger a secondary flashover, especially in load-bearing elements [35].

When subjected to fire, wood undergoes alterations in both its dimensions and geometry. Dimensional changes primarily arise from moisture alteration (drying) and thermal expansion. Meanwhile, alterations in geometry are notably influenced by the orientation of the wood fibers, with radial and tangential wood responding differently to these changes [36]. As outlined by Wang et al. (2021) [28], wood deformation occurs across four distinct stages. The initial phase involves deformation caused by shrinkage, where the wood exposed to heat from above dries more rapidly than its reverse side, resulting in the wood bending into a convex shape (∪). The subsequent phase involves deformation due to thermal expansion, occurring due to the temperature disparity between the heat-exposed surface and the opposite face, leading the wood to bend into a concave shape (∩). The third phase encompasses pyrolytic drying, wherein the exposed side undergoes significant mass loss due to pyrolytic decomposition, forming a charred layer. This phase is characterized by a convex shape (∪). The final phase involves thermal expansion attributed to the oxidation of the charred layer, emitting a substantial amount of heat. However, the formation of an ash layer hinders the dissipation of heat away from the material, complicating heat emission.

This study focuses on examining the fundamental parameters of glued laminated timber when subjected to heat flux. Understanding the material's behavior under fire conditions is essential for its utilization in contemporary structural applications. To evaluate this behavior, crucial parameters, such as the heat release rate (HRR); mass loss rate (MLR); average rate of heat emission (ARHE), along with its peak value (MARHE); time to ignition (TTI); and effective heat of combustion (EHC) were measured for combined glulam specimens composed of beech, alder, and spruce wood. These parameters serve as a foundational overview of the burning characteristics exhibited by small-scale samples of wood species combinations within a bonded structural element. They provide preliminary insights into the material's response to fire conditions before conducting full-scale tests.

## 2. Materials and Methods

### 2.1. Sample Preparation

The experimental samples were manufactured from three distinct tree species: alder (*Alnus glutinosa* (L.) Gaertn.), Norway spruce (*Picea abies* (L.) H. Karst.), and beech (*Fagus silvatica* L.). lamellae, each measuring 12 mm in thickness, were prepared from these wood species and, subsequently, adhered based on a predefined combination scheme derived from prior studies (see Figure 2). The bonding process employed two types of adhesives: a one-component polyurethane adhesive, specifically KESTOPUR 1010 (Kiilto Oy, Lempäälä, Finland), and a two-component resorcinol-phenol formaldehyde adhesive known as RPF system 1711/2520 (Akzo Nobel, Amsterdam, The Netherlands). The dimensions of the resultant test samples were 60 × 60 × 60 mm. In total, 48 samples were manufactured, accounting for eight different combinations, utilizing two types of adhesives, with three samples for each unique combination. To standardize the moisture content, all samples were conditioned within an air-conditioning chamber Clime Event 2/2000/40/3 (Weiss Umwelttechnik GmbH, Hamburg, Germany) until reaching a moisture content of 12%.

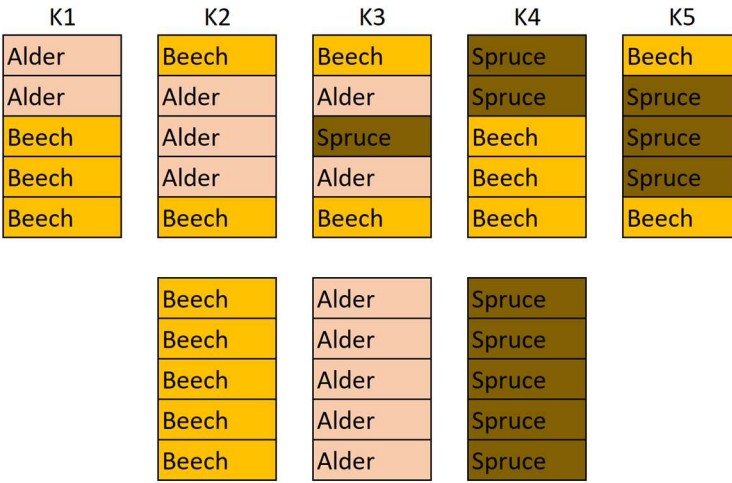

**Figure 2.** Sample combinations (Kytka et al., 2022) [37].

*2.2. Experiment Design*

The experiment was carried out using a fire test apparatus called ISO 5660-1 [38] Conical calorimeter provided by CLASSIC CZ Spol. s.r.o., Czech Republic, in a horizontal position (Figure 3) and the following characteristics were determined:

- Heat release rate (HRR) and its peak value (pHRR);
- Mass loss rate (MLR);
- Average rate of heat emission (ARHE) and its peak value (MARHE);
- Effective heat of combustion (EHC).

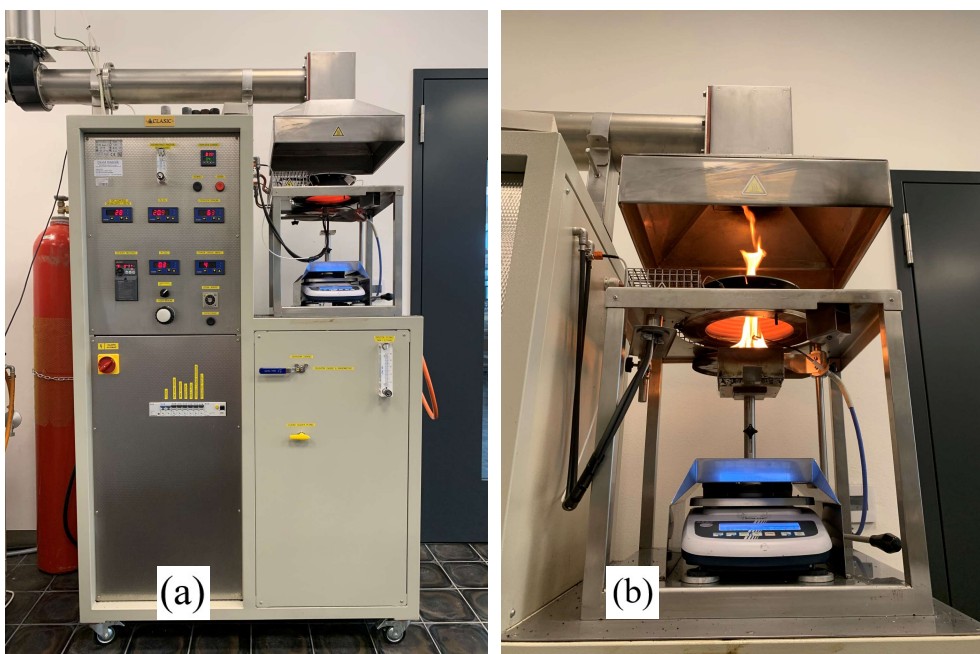

**Figure 3.** Fire test apparatus: (**a**) conical calorimeter ISO 5660-1 [38]; (**b**) sample testing.

The measured samples underwent exposure to a heat flux of 50 kW/m$^2$ for 1800 s (30 min). To ensure consistency in the area exposed to the heat flux concerning the sample size, an additional stainless-steel screen was integrated into the sample holder, adjusting the exposed area to 50 × 50 mm (see Figure 4). Each sample, enveloped in aluminum foil, was placed within the holder and positioned 30 mm away from the radiant electrical heater, without the inclusion of any supplementary grid.

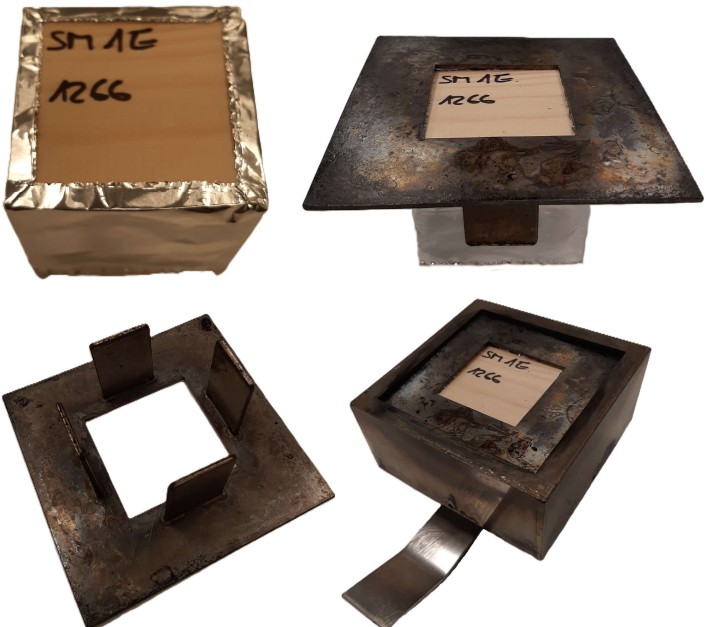

**Figure 4.** Sample in aluminum foil, stainless-steel screen, and sample holder.

Artificial ignition was facilitated using a spark plug to initiate a flame. Calibration of the calorimeter was executed following the procedures outlined in ISO 5660-1 (2015) [38], comprising three sequential steps. Initially, the accuracy of the weighing device was determined using calibrated weights provided by the equipment manufacturer. Additionally, the rate of change in weight measurement was verified. Subsequently, the second step encompassed calibrating the flow rate (0.024 ± 0.002 m³/s) and determining the oxygen fraction within the airflow (20.95 ± 0.01%). This step also involved ascertaining the C value through the utilization of a methane-burning gas burner (see Equation (1)) [38].

$$C = \frac{\dot{q}_b}{(12.54 \times 10^3)(1.10)} \sqrt{\frac{T_e}{\Delta p}} \frac{1.105 - 1.5X_{O_2}}{X_{O_2}^0 - X_{O_2}} \tag{1}$$

where $\dot{q}_b$ corresponds to the rate of heat release of the supplied methane [kW], $12.54 \times 10^3$ is the fraction of the net heat of combustion and the stoichiometric mass ratio for methane [kJ/kg], 1.10 is the ratio of the molecular weights of oxygen and air, $T_e$ is the absolute temperature of the gas at the orifice meter [K], $X_{O_2}^0$ is the initial value of the oxygen analyzer reading [-], $X_{O_2}$ is the oxygen analyzer reading mole fraction of oxygen [-], and $\Delta p$ is the orifice meter pressure differential [Pa].

The third calibration step involved ensuring the accuracy of heat flux measurements using a temperature probe to sense the radiator's temperature (848 °C), which, according to the manufacturer's specifications, had a heat flux of 50 kW.

The samples underwent exposure to heat flux in two different orientations, both with the heat flux directed horizontally from above the sample. However, the orientation of the sample concerning the applied heat flux varied, as depicted in Figure 5. These distinct orientations aimed to depict potential heat flux scenarios from a hypothetical fire that poses a threat to the structure. Owing to the conical design of the calorimeter, conducting tests with laterally applied heat flux was not feasible. Nevertheless, the obtained results offer valuable insights into the behavior of the bonded joint when exposed to the heat flux, despite the limitations in the test configuration. The third step was to calibrate the accuracy of the heat flux measurements using a temperature probe that sensed the temperature of the radiator (848 °C) with, according to the manufacturer's instructions, a heat flux of 50 kW.

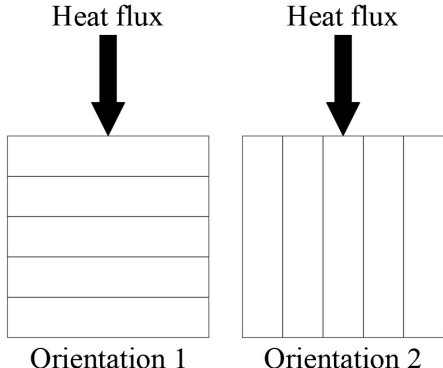

**Figure 5.** Sample orientations.

*2.3. Calculations*

Heat release rate (HRR) was calculated according to Equation (2) (ISO 5660-1, 2015) [38].

$$HRR_A(t) = \frac{(\Delta h_c/r_0) \times (1.10) \times C \times \sqrt{\frac{\Delta p}{T_e}} \times \frac{X_{O_2}^0 - X_{O_2}}{1.105 - 1.5X_{O_2}}}{A_s} \tag{2}$$

where $HRRA(t)$ is the heat release rate per period and unit area [kW/m$^2$], $\Delta h_c$ is the net heat of combustion [kJ/g], $r_0$ is the stoichiometric oxygen/fuel mass ratio [-] (according to ISO 5660-1 (2015) [38], the ratio $(\Delta h_c/r_0) = 13.1 \times 10^3$ kJ/kg can be used), $C$ is the orifice flow meter calibration constant [m$^{1/2}$ g$^{1/2}$ K$^{1/2}$], $\Delta p$ is the orifice meter pressure differential [Pa], $T_e$ is the absolute temperature of the gas at the orifice meter [K], $X_{O_2}^0$ is the initial value of the oxygen analyzer reading [-], $X_{O_2}$ is the oxygen analyzer reading mole fraction of oxygen [-], and $A_s$ is the initially exposed surface area of the specimen [m$^2$].

The calculation of the mass loss rate (MLR) was performed following Equations (3)–(7) [38], divided into at least five distinct steps. The initial two equations (Equations (3) and (4)) were used to account for the onset of mass loss, while the final two equations (Equations (6) and (7)) were employed to consider the terminal stage within the quasi-static phase of the combustion process. Equation (5) was applicable for any measurement within the range of $1 < i < n—1$, where "$n$" represents the total number of measurements conducted during the experiment.

$$-[MLR]_{i=0} = \frac{25m_0 - 48m_1 + 36m_2 - 16m_3 + 3m_4}{12\Delta t} \tag{3}$$

$$-[MLR]_{i=1} = \frac{3m_0 + 10m_1 - 18m_2 + 6m_3 - m_4}{12\Delta t} \tag{4}$$

$$-[MLR]_i = \frac{-m_{i-2} + 8m_{i-1} - 8m_{i+1} + m_{i+2}}{12\Delta t} \tag{5}$$

$$-[MLR]_{i=n-1} = \frac{-3m_n - 10m_{n-1} + 18m_{n-2} - 6m_{n-3} + m_{n-4}}{12\Delta t} \tag{6}$$

$$-[MLR]_{i=n} = \frac{-25m_n + 48m_{n-1} - 36m_{n-2} + 16m_{n-3} - 3m_{n-4}}{12\Delta t} \tag{7}$$

where $MLR$ is the mass loss rate of sample per scan [g/s], $m$ is the mass of specimen per scan [g], $n$ is the number of measurements, and $t$ is the time [s].

The average rate of heat emission (ARHE) serves as a metric for assessing the distribution of accumulated radiation heat over time, providing insight into its gradual release. Meanwhile, the maximum average rate of heat emission (MARHE) signifies a material's

susceptibility to fire and the rapidity with which it emits heat. The calculation for MARHE is performed according to Equation (8) [39].

$$ARHE(t_n) = \frac{\sum_{2}^{n}\left((t_n - t_{n-1}) \cdot \frac{q_n + q_{n-1}}{2}\right)}{t_n - t_0} \tag{8}$$

where $ARHE(t_n)$ is the average rate of heat emission [kW/m$^2$], $t_n$ is the time [s], and $q_n$ is the heat release rate at measured time [kW/m$^2$].

The effective heat of combustion (EHC) denotes the quantity of energy released during the combustion process in relation to the material's lost mass. This parameter is notably impacted by the degradation method employed and the moisture content inherent in the material. Notably, for lignocellulosic materials that possess diverse degradation pathways, the EHC is not a constant value. The calculation for EHC is outlined in Equation (9) [38].

$$EHC = \frac{HRR}{MLR} \tag{9}$$

where EHC is the effective heat of combustion [MJ/kg], HRR is the heat release rate [kW/m$^2$], and MLR is the mass loss rate [g].

Tukey's test, which was implemented through Statistica 14 by TIBCO Software Inc., Palo Alto, CA, USA, was employed to ascertain the statistical significance of disparities between the measured combinations. Furthermore, all graphs presented in the results underwent a smoothing process utilizing a 5-point moving average. It is worth noting that this smoothing technique, as stated by Morrisset et al. (2022) [19], introduces a maximum distortion to the data of 2%.

## 3. Results and Discussion

The outcomes from the fire test illustrating the heat release rate are depicted in Figure 6, while the mass loss rate results are represented in Figure 7. These graphs showcase an initial rapid escalation, followed by a subsequent decline, which is a characteristic trend observed in wood combustion behavior. The decline primarily arose from the initial formation of a charred layer, serving as an insulating barrier against further heat flux. Beyond the 200th second, the behavior of the individual combinations began to diverge. However, a common feature among most combinations was the emergence of a secondary peak, which was primarily attributed to surface destruction and the development of new cracks, enabling the escape of newly formed gaseous products from pyrolysis.

Among the homogeneous combinations (beech, spruce, alder), beech wood demonstrated the most pronounced response to this process, while spruce wood exhibited the least pronounced response. Comparable heat release rate trends were also observed in the research conducted by Repič et al. (2023) [40], who specifically investigated the fire properties related to different treatments of beech wood through mineralization. However, in the case of untreated beech wood, the development of the second peak is notably substantial, with the heat release rate reaching up to 200% of the first peak. Conversely, findings from Martinka et al. (2014) [41], which focused on spruce wood, indicate that spruce wood showcases the highest heat release rate values at the first peak, with the second peak reaching only half of the maximum values observed.

This decomposition resulted from the conversion of larger molecules—cellulose, hemicelluloses, and lignin—into smaller molecules, which acted as catalysts, instigating the degradation of other wood components [42]. Following the combustion of the beech layer, the pyrolysis process was initiated within the inner lamellae, leading to the decomposition of its hemicelluloses and volatiles, releasing flammable gases, corresponding to the highest levels of released heat. Notably, as the uppermost lamellae originated from the same material, the extent of hemicellulose decomposition and the emission of flammable gases remained consistent, sustaining a constant supply of this fuel source for nearly 4 min. As the heat released over time progressed, it approached values akin to those observed in

a homogeneous combination for the specified wood species. Nonetheless, a process of glowing combustion persisted within the charred beech layer, further contributing to heat emission in the combustion process [43].

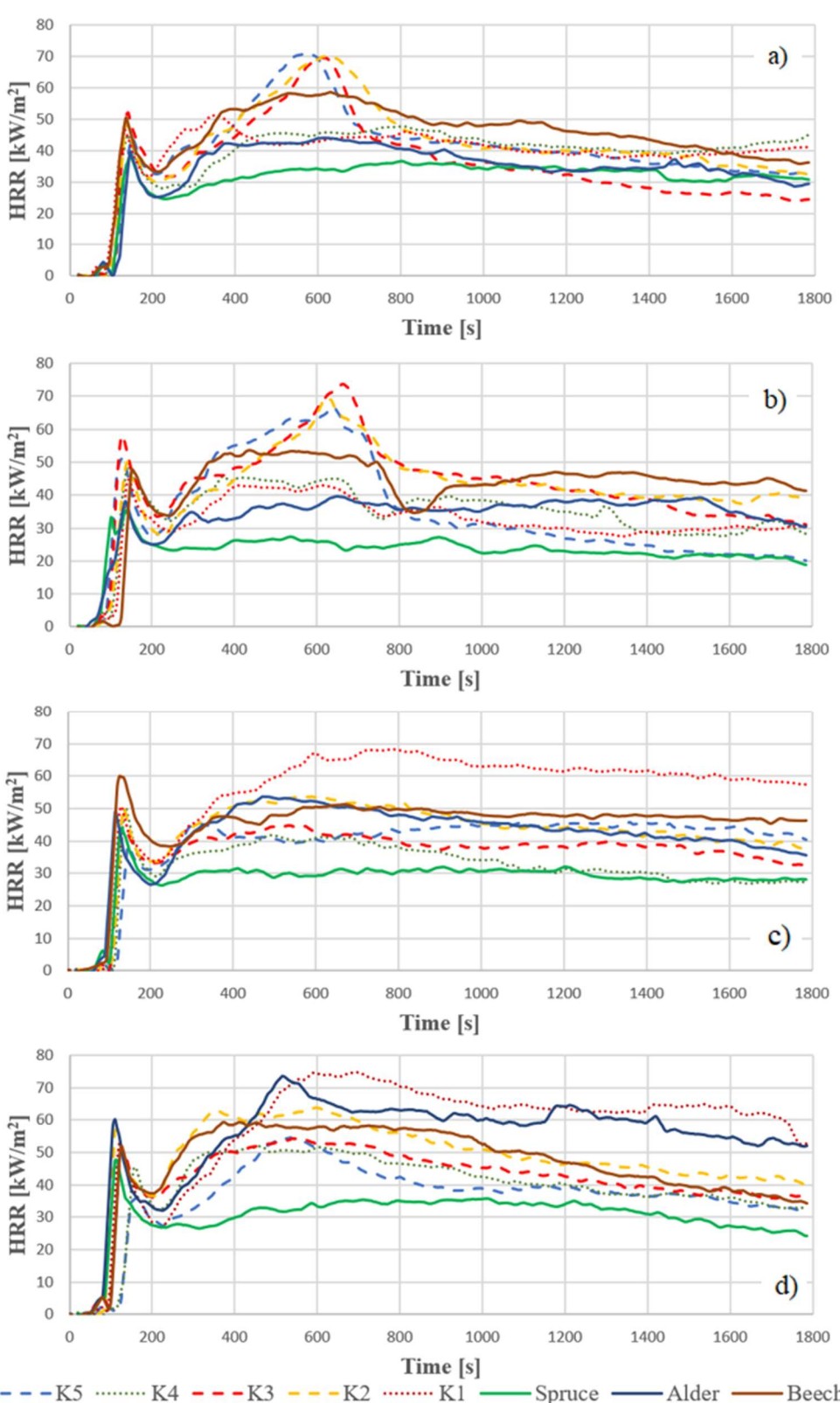

**Figure 6.** Heat release rate (HRR) for glulam combinations: (**a**) glued using FPR, orientation 1; (**b**) glued using PUR, orientation 1; (**c**) glued using FPR, orientation 2; (**d**) glued using PUR, orientation 2.

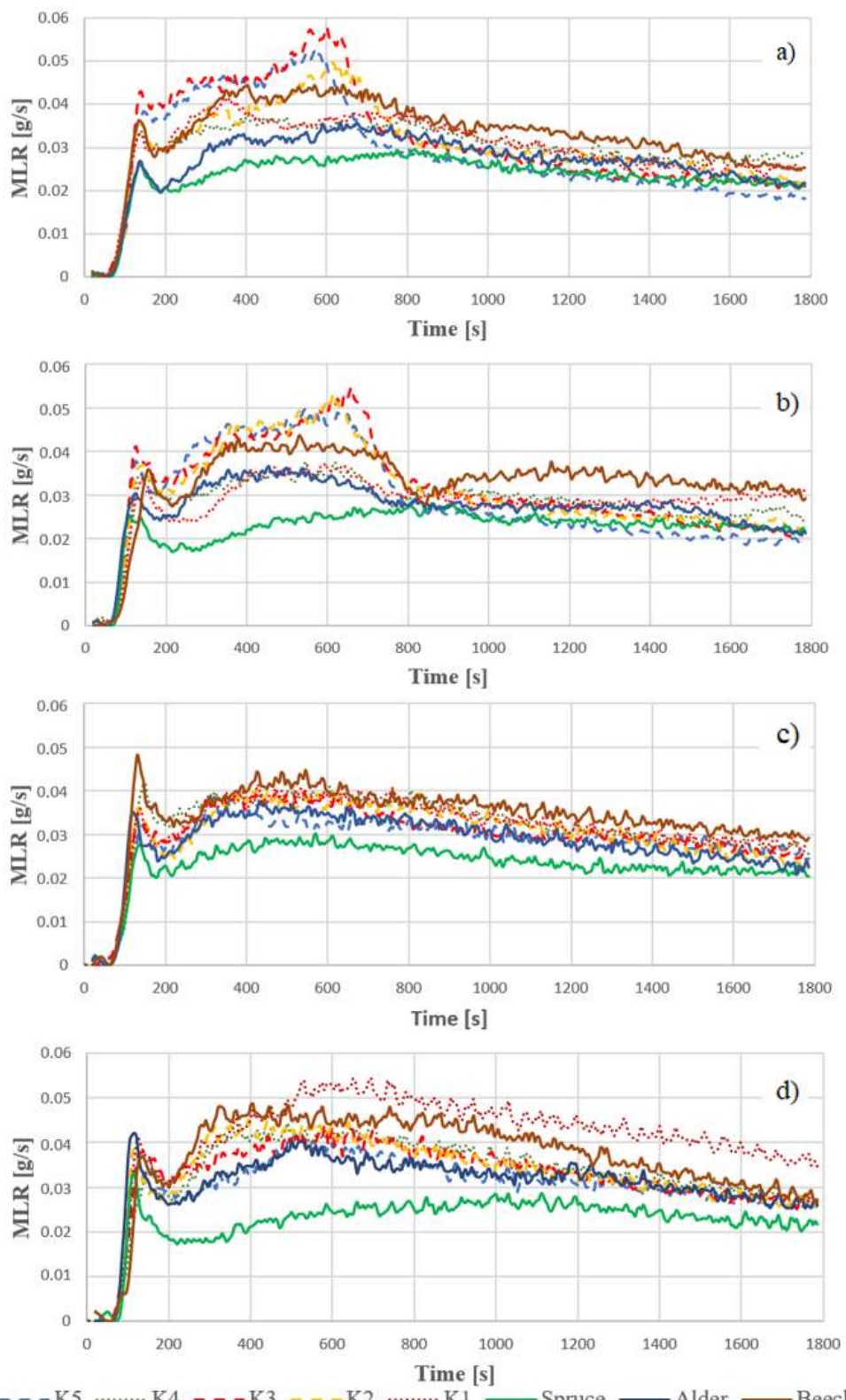

**Figure 7.** Mass loss rate (MLR) for glulam combinations: (**a**) glued using FPR, orientation 1; (**b**) glued using PUR, orientation 1; (**c**) glued using FPR, orientation 2; (**d**) glued using PUR, orientation 2.

In inhomogeneous symmetrical combinations, the emergence of the second peak was distinguished by a sharp surge in the released heat, primarily due to the thermal decomposition of the surface layer of beech wood (Figure 8).

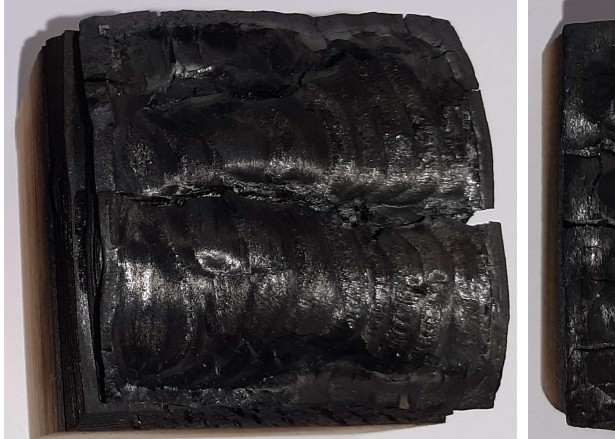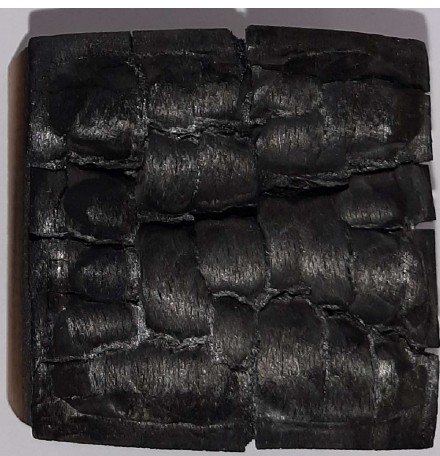

**Figure 8.** Fissured top beech lamella at K5 RPF (**left**) and K2 PUR (**right**).

Regarding the inhomogeneous asymmetric combinations (K1 and K4), the rise in the second peak was more gradual, aligning closely with the wood utilized in the upper layers. However, the heat release rate (HRR) began to escalate as time progressed due to the decomposition of the beech wood. This discrepancy in burning behaviors between softwood and hardwood species became noticeable, particularly showcasing a faster escalation in the second peak for softwood species [44].

The observed differences in burning behaviors are attributed, according to Shapchenkova et al. (2022) [45], to the varying chemical compositions inherent in different wood species. As per their findings, wood decomposition comprises three distinct stages. In the initial stage, the decomposition of hemicelluloses and volatile extractives predominates, which is a phase where beech wood holds dominance due to its higher hemicellulose content. This dominance notably influences the escalation in released heat observed in homogeneous and symmetrical inhomogeneous combinations. The second phase involved the decomposition of cellulose, where beech wood exhibited lower thermal stability of cellulose compared with alder wood. The third stage involves the decomposition of lignin, where, again, alder demonstrated higher thermal stability than beech wood.

In contrast, Richter et al. (2019) [46] suggested that softwoods contain higher proportions of lignin, cellulose, and extractives, particularly resins, compared with hardwoods. This composition influences the initial stage of pyrolysis, where extractives decompose. However, subsequent to the decomposition of these wood components, a steadier progression of decomposition was observed, primarily due to the higher thermal stability of cellulose and lignin present in spruce wood.

In the case of orientation 2, wherein the heat flux was directed toward the bonded edges, the observed pattern appeared to be analogous but more consistent compared with when the heat flux was applied solely to the top surface of the sample. The adhesive type utilized played a significant role, notably influencing the initial heat release rate (HRR) values. The PUR adhesive exhibited higher initial HRR values, primarily due to delamination of the bonded joint, while the samples bonded with RPF adhesive were more prone to wood cracking than within the bonded joint. This effect was particularly prominent, especially in the combinations involving alder wood, where almost the entire bonded joint of the specimen experienced delamination (Figure 9).

Furthermore, findings from Yang et al. (2009) [47] indicate that glulam bonded with resorcinol adhesive, when exposed to heat flux from the lateral side, exhibits lesser charring compared with exposure from the surface, aligning consistently with the distinctions observed between Figure 6c,d in our case.

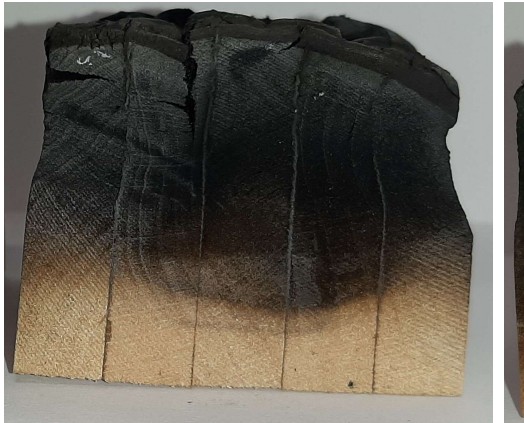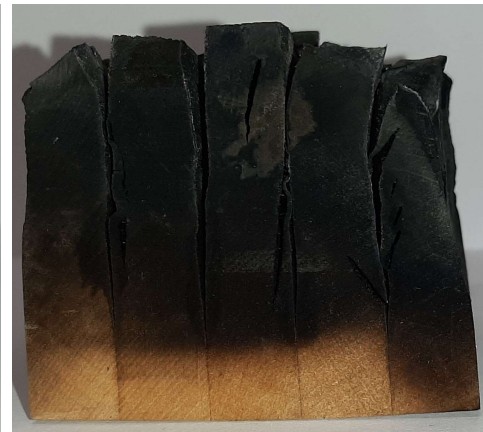

**Figure 9.** Alder bonded using RPF (**left**) and PUR (**right**) (orientation 2).

In wood bonding, the choice of adhesive is a pivotal consideration, with the thermal stability of adhesives serving as a primary determinant of bond strength. The thermal stability of adhesives stands as a pivotal attribute affecting bond strength, especially under varying temperature conditions. Polyurethane (PUR) adhesives can exhibit commendable thermal stability, particularly when subjected to moderate temperatures. According to findings by Na et al. (2005) [48], the thermal stability of PUR adhesives is predominantly reliant on the adhesive composition, specifically emphasizing the crucial role of the NCO/OH ratio. When encountering fire and high temperatures, PUR adhesives generally demonstrate favorable fire resistance properties in comparison with various other adhesive types [49]. However, under applied heat flux, PUR adhesives might undergo softening or experience a reduction in bond strength.

Contrarily, resorcinol-phenol formaldehyde (RPF) adhesives showcase notable high thermal stability, capable of enduring elevated temperatures compared with many other adhesive varieties. These adhesives retain their bond strength and structural integrity, even under significant heat exposure [50]. RPF adhesives exhibit robust resistance to softening, melting, or delamination when subjected to high temperatures, rendering them suitable for applications where fire resistance and heat stability are critical [51]. In a comparative study by Hartig and Haller (2023) [52] involving gluing spruce, beech, and poplar wood using PUR and phenol formaldehyde glue, poplar wood, closely aligned in density to alder wood, exhibited a similar burning pattern to alder wood, with the second peak reaching heat release rate (HRR) values akin to the first peak.

Delamination within the bonded joint poses a significant challenge, particularly in the scenario of heat flux directed from top to surface (orientation 1). In instances involving RPF-bonded samples, partial delamination of the first bonded joint was observed and always limited to a maximum of 25% of the bonded joint area. Conversely, cracks within the wood were more prevalent, notably perpendicular to the fibers (Figure 8), as supported by findings from Li et al. (2017) [53]. For PUR adhesive, a notable trend observed was the complete delamination of the first two bonded joints along the direction of heat flow (Figure 10). This led to the exposure of the virgin material, which further contributed to its pyrolysis in the combustion process.

Moreover, PUR adhesives demonstrate an accelerated charring rate and mass loss rate compared with RPF adhesives [54]. The progression of delamination was most pronounced in beech (Figure 6b), where after 800 s, the surface of the initial bonded joint experienced total delamination, initiating the pyrolysis of the subsequent layer. Emberley et al. (2017) [9] also investigated the effect of delamination on cross-laminated spruce wood, finding a rapid surge in the heat release rate (HRR) subsequent to the delamination of the first joint and exposure of the untouched material in the second joint.

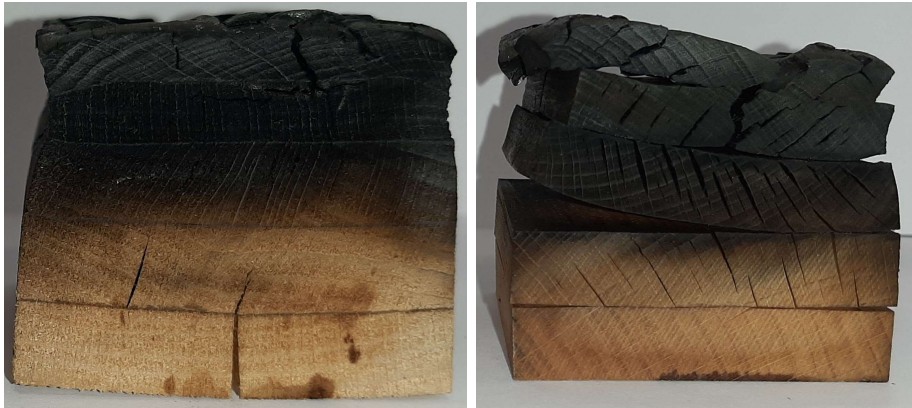

**Figure 10.** Beech sample bonded using RPF (**left**) and PUR (**right**).

A holistic evaluation of the material's fire performance necessitates consideration of additional characteristics, as outlined in Table 2. Notably, all combinations exhibited a consistent ignition, with a steady flame observed between the first and second minute following the initiation of heat flux exposure. The analysis revealed no statistically significant difference in time to ignition (TTI) across various combinations and sample orientations. This uniformity in TTI can be attributed to its direct correlation with the heat flux increase rate (HFIR) [55].

The peak values of the heat release rate (pHRR) followed a comparable trend, showcasing no statistically significant difference between the adhesives utilized. However, the orientation of the sample notably influenced the magnitude of heat released, which was particularly evident in the case of alder and combinations incorporating alder. Wood density played a pivotal role here, as a lower density enhanced the porosity, resulting in reduced thermal conductivity and the emergence of localized heat accumulation, thus promoting increased flame spread [7]. The timing of reaching the peak HRR demonstrated two distinct intervals. The initial interval, occurring around the 120th second, marked the onset of charred layer formation. Subsequently, around the 600th second, a secondary interval emerged characterized by the creation of new fissures from which pyrolysis gases emanated, further sustaining the combustion process. These intervals signify critical stages in the evolving dynamics of heat release during the combustion process.

The homogeneous combination of spruce wood bonded with RPF adhesive, according to our findings, displayed the lowest heat release rate (HRR) and mass loss rate (MLR). The lower flammability observed in this combination, as indicated by its low HRR and MLR, can be attributed to the rapid formation of a charred layer. This charred layer effectively acted as insulation, limiting further penetration of heat into the material.

Compared with other combinations primarily bonded with PUR adhesive, the spruce wood combined with RPF adhesive did not exhibit delamination but rather showed cracking within the wood. Additionally, its relatively low maximum HRR (pHRR) and the quick increase in HRR within a short period (time to ignition around 90 s), followed by a swift decrease and stabilization of HRR at around 30 kW/m$^2$, made this combination advantageous from a fire safety perspective. This conclusion aligns with the prevalent use of softwood in the production of glulam, further validating the favorable fire resistance characteristics observed in the homogeneous combination of spruce wood bonded with RPF adhesive.

In a global context, while spruce wood presents advantageous fire-resistant properties in glulam structures, it might not remain the most preferable option for various reasons. One primary factor is the ongoing shift away from spruce in favor of other tree species, particularly hardwoods, which are becoming more favored. Global climate change also contributes to the decline of spruce [56]. Additionally, considering mechanical load capacities, several other commonly available wood species demonstrate superior mechanical

properties compared with spruce. For instance, when combining spruce with beech wood (as in combination K5), our previous research has revealed an approximate 10% increase in both the modulus of elasticity (MOE) and bending strength (MOR) when using PUR adhesive [37]. Furthermore, with RPF adhesive, this combination approached the mechanical properties of a homogeneous beech wood combination, although these specific findings have not been published yet. This suggests that combining spruce with certain hardwoods, such as beech, resulted in improved mechanical characteristics, especially when considering bonding with different adhesives. These mechanical enhancements offer an alternative option for structural applications compared with spruce-based glulam structures.

**Table 2.** Burning properties of glued combinations.

| Combination-Adhesive-Orientation | TTI [s] | pHRR [kW/m$^2$] | Time at pHRR [s] | EHC [MJ/kg] | MARHE [kW/m$^2$] |
|---|---|---|---|---|---|
| Beech-PUR-1 | 123 | 53.6 | 425 | 3.2 | 42.6 |
| Beech-PUR-2 | 99 | 59.2 | 400 | 3.0 | 49.4 |
| Beech-RPF-1 | 95 | 58.6 | 620 | 3.2 | 46.0 |
| Beech-RPF-2 | 118 | 60.0 | 115 | 3.2 | 41.6 |
| Alder-PUR-1 | 88 | 39.5 | 640 | 3.1 | 34.3 |
| Alder-PUR-2 | 74 | 73.4 | 505 | 4.4 | 55.1 |
| Alder-RPF-1 | 113 | 43.9 | 605 | 3.1 | 34.6 |
| Alder-RPF-2 | 91 | 53.5 | 470 | 3.6 | 42.4 |
| Spruce-PUR-1 | 93 | 56.6 | 125 | 2.5 | 25.2 |
| Spruce-PUR-2 | 80 | 47.5 | 105 | 3.0 | 31.1 |
| Spruce-RPF-1 | 107 | 38.1 | 135 | 3.3 | 30.9 |
| Spruce-RPF-2 | 103 | 44.2 | 120 | 3.1 | 28.5 |
| K1-PUR-1 | 106 | 48.9 | 130 | 2.6 | 34.2 |
| K1-PUR-2 | 92 | 74.6 | 685 | 3.4 | 57.7 |
| K1-RPF-1 | 95 | 51.1 | 345 | 3.3 | 39.8 |
| K1-RPF-2 | 96 | 68.4 | 770 | 4.4 | 55.8 |
| K2-PUR-1 | 98 | 69.5 | 610 | 3.5 | 43.6 |
| K2-PUR-2 | 83 | 63.9 | 585 | 3.6 | 51.0 |
| K2-RPF-1 | 104 | 70.1 | 615 | 3.4 | 44.9 |
| K2-RPF-2 | 101 | 53.7 | 580 | 3.5 | 43.2 |
| K3-PUR-1 | 92 | 73.7 | 650 | 3.4 | 46.7 |
| K3-PUR-2 | 99 | 54.4 | 530 | 3.1 | 43.9 |
| K3-RPF-1 | 103 | 69.5 | 600 | 2.8 | 41.9 |
| K3-RPF-2 | 104 | 44.8 | 525 | 3.0 | 36.9 |
| K4-PUR-1 | 108 | 49.7 | 130 | 2.9 | 36.6 |
| K4-PUR-2 | 121 | 52.0 | 450 | 2.9 | 41.0 |
| K4-RPF-1 | 97 | 47.8 | 800 | 3.4 | 39.8 |
| K4-RPF-2 | 110 | 50.4 | 130 | 2.4 | 33.8 |
| K5-PUR-1 | 91 | 66.6 | 630 | 2.9 | 45.6 |
| K5-PUR-2 | 125 | 54.5 | 525 | 3.0 | 37.1 |
| K5-RPF-1 | 108 | 70.8 | 570 | 3.5 | 43.6 |
| K5-RPF-2 | 109 | 46.1 | 305 | 3.5 | 39.8 |

Note: TTI is the time to ignition [s], pHRR is the maximum value of heat release rate [kW/m$^2$], EHC is the effective heat of combustion [MJ/kg], and MARHE is the maximum value of average rate of heat emission [kW/m$^2$].

Also, the experimental conditions in this study represented an extreme scenario, subjecting the glulam timber to intense heat flux without any protective treatment. In real-world applications, wooden load-bearing elements are often treated with fire-retardant materials or covered with non-combustible layers, such as plasterboards. This approach significantly enhances fire resistance and safety. Finding a balance between various factors—like cost-effectiveness, mechanical strength, environmental impact, technical requirements, manufacturing feasibility, and fire performance—is crucial in designing and implementing wooden structures. This balance ensures both structural integrity and compliance with

safety standards, allowing for optimal performance in the event of a fire while considering practical and economic aspects.

The outcomes presented in this paper hold practical implications for the design and assessment of fire-resistant wooden load-bearing elements. According to Babrauskas and Peacock (1992) [57], parameters like the heat release rate (HRR) and its associated mass loss rate (MLR) serve as fundamental characteristics to gauge the flammability of materials within a structure. These parameters are pivotal not only in assessing flammability but also in predicting critical fire dynamics, such as the likelihood of flashover and smoke production rates [58]. In high-rise structures, particularly where the airflow crucially influences combustion, comprehending the rate of heat release becomes paramount, particularly concerning upper-floor openings. Controlling the rate at which the supporting structure burns is vital for averting building collapse and minimizing potential fatalities, as indicated by prior studies [59]. Fire engineers rely on such factors to model and design load-bearing elements, emphasizing the importance of considering combustion behavior in structural design and fire safety [47,60,61]. Incorporating these insights can significantly enhance the safety and resilience of wooden load-bearing structures against fire incidents.

## 4. Conclusions

From this investigation that involved five-layer glulams bonded with PUR and RPF adhesives made from a blend of alder, spruce, and beech wood exposed to a 50 kW/m$^2$ heat flux, several conclusions can be drawn from the results:

Heat release rate (HRR) and mass loss rate (MLR): All combinations experienced a rapid rise in HRR and MLR within the initial 1–2 min of heat exposure, followed by a decline due to the formation of a charring layer. A subsequent increase in HRR and MLR was observed when the charred layer fissured, creating a second peak. Combinations with beech wood in the upper lamella (K2, K3, and K5) exhibited the highest HRR and MLR values. Among homogeneous combinations, beech wood glulam demonstrated the highest HRR and MLR, while spruce wood glulam exhibited the lowest.

Time to ignition (TTI): the time to ignition ranged from 74 to 125 s across all combinations and did not exhibit statistically significant differences.

Effect of adhesives: The behavior of the glulam under heat flux was influenced by the type of adhesive used. PUR adhesive showcased total delamination in the first two glued joints, which was notably evident in the homogeneous beech combination. Conversely, RPF adhesive displayed delamination limited to the first glued joint, up to a maximum of 25% of the glued joint area.

Effective heat of combustion (EHC): no statistically significant differences were observed in the effective heat of combustion between wood species or combinations, except for glulams from alder wood and combination K1, which surpassed values of 4 MJ/kg when the heat flux was oriented from above toward the glued edge.

Maximum average rate of heat emission: The MARHE, which is indicative of flammability, was the highest for homogeneous beech wood glulam and the lowest for spruce wood glulam. Combined beams reflected the proportional representation of each wood species in the combination.

These findings underscore the distinctive behavior of different wood species and combinations when subjected to heat flux, highlighting the significant impact of wood species, adhesive type, and orientation on the combustion characteristics, such as HRR, MLR, ignition time, and flammability.

**Author Contributions:** Conceptualization, T.K., M.G. and E.K.; methodology, T.K. and M.G.; software, T.K., L.S. and D.N.; validation, T.K., L.S. and D.N.; formal analysis, L.S., D.N. and S.D.; investigation T.K., M.G. and E.K.; resources, T.K., L.S., D.N. and S.D.; data curation, T.K., L.S., D.N. and E.K.; writing—original draft preparation, T.K., M.G. and E.K.; writing—review and editing, T.K., M.G., L.S., D.N. and E.K.; visualization, L.S., D.N. and S.D.; supervision, T.K., M.G. and E.K.; project administration, T.K. and M.G.; funding acquisition, T.K. and M.G. All authors have read and agreed to the published version of the manuscript.

**Funding:** This research was funded by the Internal Grant Agency (IGA) of the Faculty of Forestry and Wood Sciences (Project No. IGA A_21_23) of the Czech University of Life Sciences, Czech Republic. Also, the research was funded by "Advanced research supporting the forestry and wood-processing sector's adaptation to global change and the 4th industrial revolution", no. CZ.02.1.01/0.0/0.0/16_019/0000803, financed by OP RDE, The Ministry of Education, Youth and Sports, of the Czech Republic.

**Data Availability Statement:** Data available on personal request to correspondence author.

**Conflicts of Interest:** The authors declare no conflicts of interest.

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
