# Peer review of "Burning Properties of Combined Glued Laminated Timber"

_fire, doi:10.3390/fire7010030_

Round 1

Reviewer 1 Report

Comments and Suggestions for Authors

This manuscript presents a study on the burning properties of Glulam with several combinations with different wood species and adhesives. The manuscript could be accepted if the following problems/ questions can be addressed:

1. Abstract should be revised to added some suggestions for selecting the combination.

2. The paper finished without a conclusion?Is the Discussion should be conclusion?

3. Could the authors provide the optimized suggestions of combinations when subjected to fire?

4. The thermal material properties (such as density, thermal conductivity, specific heat capacity) are missing... The authors should added them as possible, since they are very important to evaluate the thermal performance of wood in fire (https://doi.org/10.1016/j.tws.2023.111240).

5. Could the authors add some pictures of the experiment and the detailed instruments and set-up?

6. Have the authors measured the temperatures?

Author Response

  1. Abstract should be revised to added some suggestions for selecting the combination.

Explanation:

The selection of tree species in this study was purposeful and aimed to represent diverse characteristics. Spruce, being a common softwood species and widely abundant in the Czech Republic, was chosen to stand as a representative species in this category. Beech, a high-density hardwood species, was selected as a typical example of hardwood varieties prevalent in the area. Alder, being a low-density hardwood species but possessing relatively favourable mechanical properties compared to other common options like poplar, was included in the study. This choice was motivated by the desire to explore wood species with varying mechanical strengths and densities. The combinations of these species were specifically chosen based on their mechanical behaviour under bending stresses, which typically induce high tensile and compressive stresses in the outer layers and elevated shear stresses in the central layer. Hence, the arrangement of stronger lamellae in the outer layers or the central lamella was deliberate. Weaker lamellae were placed in the middle layers since they are not as exposed to such high forces. Overall, the objective of this research was to expand the understanding of the properties exhibited by composite bonded elements. The study aimed to focus on their applicability in horizontal structures that might be exposed to fire, thereby contributing to the broader knowledge and utilization of these materials in such scenarios.

Added short text in abstract.

  1. The paper finished without a conclusion?Is the “Discussion” should be “conclusion”?

Yes, chapter “Discussion” has to be “Conclusion”. Corrected.

  1. Could the authors provide the optimized suggestions of combinations when subjected to fire?

Added text in discussion after table 3, source 56.

  1. The thermal material properties (such as density, thermal conductivity, specific heat capacity) are missing... The authors should added them as possible, since they are very important to evaluate the thermal performance of wood in fire (https://doi.org/10.1016/j.tws.2023.111240).

Added table in introduction, sources 13 and 16-18.

  1. Could the authors add some pictures of the experiment and the detailed instruments and set-up?

Changed Figure 4, added Fig. 5 and 9-11.

  1. Have the authors measured the temperatures?

The temperature was measured only on the calorimeter itself to ensure heat flux. The temperature inside the sample was not measured for example by sensors.

Reviewer 2 Report

Comments and Suggestions for Authors

fire-2781622

Burning properties of combined glued laminated timber

Ø  The introduction provides a clear overview of the focus on burning properties, but it would be beneficial to include a brief rationale for studying these properties in combined glulams bonded by PUR and RPF adhesives.

Ø  Provide more details on the experimental setup, such as the testing apparatus used for measuring heat release rate, mass loss rate, and other parameters.

Ø  Explain the significance of the observed changes in heat release, mass loss, and other parameters in the context of fire safety and structural implications.

Ø  Elaborate on the reasons behind the observed differences in burning properties among spruce, alder, and beech wood. Discuss the inherent characteristics of each wood species that contribute to variations in heat release and combustion behavior.

Ø  Further explore and discuss the differences in burning properties between PUR and RPF adhesives. Provide insights into the thermal stability and performance of each adhesive under the applied heat flux.

Ø  Discuss the practical implications of the findings for the use of combined glulams in real-world applications. How might the observed differences in burning properties influence the structural integrity and fire safety of these materials?

Comments on the Quality of English Language

The linguistic level and the mechanics of English writing are not appropriate for publication. There are few grammatical and typing errors in the manuscript, so please check and revise. The way of writing is not clear and it is difficult for the readers to understand. The paper should be rewritten and proofread again thoroughly. Extensive editing of English language is required.

Author Response

1. The introduction provides a clear overview of the focus on burning properties, but it would be beneficial to include a brief rationale for studying these properties in combined glulams bonded by PUR and RPF adhesives.

Added text in introduction, sources 30-34.

2. Provide more details on the experimental setup, such as the testing apparatus used for measuring heat release rate, mass loss rate, and other parameters.

Added text in Experimental design.

3. Explain the significance of the observed changes in heat release, mass loss, and other parameters in the context of fire safety and structural implications.

Added text in introduction, sources 20-23.

4. Elaborate on the reasons behind the observed differences in burning properties among spruce, alder, and beech wood. Discuss the inherent characteristics of each wood species that contribute to variations in heat release and combustion behaviour.

Added text in discussion, sources 45 a 46.

5. Further explore and discuss the differences in burning properties between PUR and RPF adhesives. Provide insights into the thermal stability and performance of each adhesive under the applied heat flux.

Added discussion, sources 48 -51.

6. Discuss the practical implications of the findings for the use of combined glulams in real-world applications. How might the observed differences in burning properties influence the structural integrity and fire safety of these materials?

Added discussion, sources 57-61.

Comments on the Quality of English Language

The linguistic level and the mechanics of English writing are not appropriate for publication. There are few grammatical and typing errors in the manuscript, so please check and revise. The way of writing is not clear and it is difficult for the readers to understand. The paper should be rewritten and proofread again thoroughly. Extensive editing of English language is required.

Whole article was rewritten.

Round 2

Reviewer 1 Report

Comments and Suggestions for Authors

The authors have addressed all the questions. Now this manuscript can be published.